# Major dietary patterns of community dwelling adults and their associations with impaired blood glucose and central obesity in Eastern Ethiopia: Diet-disease epidemiological study

Berhanu Abebaw Mekonnen[1], Abdu Oumer[2]*, Ahmed Ale[3], Aragaw Hamza[4], Imam Dagne[2], Abdurezak Adem Umer[2], Dilnessa Fentie[3], Muluken Yigezu[2], Zerihun Tariku[2], Shambel Abate[2]

1 Department of Nutrition and Dietetics, School of Public Health, Bahir Dar University, Bahir Dar, Ethiopia, 2 Department of Public Health, College of Medicine and Health Sciences, Dire Dawa University, Dire Dawa, Ethiopia, 3 School of Medicine, College of Medicine and Health Sciences, Dire Dawa University, Dire Dawa, Ethiopia, 4 Department of Anesthesia, College of Medicine and Health Sciences, Dire Dawa University, Dire Dawa, Ethiopia

* omab2320@gmail.com

## Abstract

### Backgrounds

Unhealthy dietary intake is an important preventable risk factor for obesity and impaired blood glucose (IBG), ultimately increasing the risk of non-communicable diseases. When compared to individual food intakes, dietary patterns are a stronger predictor of health outcomes and should be systematically evaluated where such evidence is lacking. This study evaluated dietary patterns and their association with the risk of central obesity and IBG among adults.

### Methods

A community-based survey was conducted among 501 randomly-selected adults from Eastern Ethiopia. Data was collected using a semi-structured questionnaire during a face-to-face interview that included sociodemographic and lifestyle factors, as well as a validated 89-item food frequency questionnaire (collected over one month). Principal component analysis was used to derive the dietary pattern. While central obesity was assessed using waist and/or hip circumference measurements, fasting blood sugar was used for IBG. A multivariable logistic regression model was fitted with an odds ratio, 95% confidence intervals, and p-values reported.

### Results

A total of 501 adults (95.3%) were interviewed, with a mean age of 41 years (±12). Five major dietary patterns explaining 71% of the total variance were identified: "nutrient-dense foods", "high fat and protein", "processed foods", "alcohol drinks", and "cereal diets". While 20.4% (17.0–24.2%) had IBG, 14.6% (11.8–17.9) were centrally obese, and 94.6% (92.3–

**Data Availability Statement:** All relevant data are included within the paper and its Supporting Information files.

**Funding:** This study was funded by Dire Dawa University, under annual competitive research grant. The funders had no role in study design, data collection and analysis, the decision to publish, or the preparation of the manuscript.

**Competing interests:** The authors have declared that no competing interests exist.

**Abbreviations:** CI, Confidence Interval; DP, Dietary Patterns; FFQ, Food Frequency Questionnaire; GPAQ, General Physical Activity Questionnaire; IBG, Impaired Blood Glucose; HC, Hip Circumference; K, Sampling Interval; METS, Metabolic Equivalents; PCA, Principal Component Analysis; NCDs, Non communicable Diseases; SD, Standard Deviations; VIF, Variance Inflation Factor; WC, Waist Circumference; WHCR, Waist–to–Hip Circumference Ratio; WHO, World Health Organization.

96.3) had an increased waist-to-hip circumference ratio. Central obesity is associated with upper wealth status (AOR = 6.92; 2.91–16.5), physical inactivity (AOR = 21.1; 2.77–161.4), a diet high in nutrient-dense foods (AOR = 1.75; 0.75–4.06), processed foods (AOR = 1.41; 0.57–3.48), and cereal diets (AOR = 4.06; 1.87–8.82). The burden of IBG was associated with upper wealth status (AOR = 2.36; 1.36–4.10), physical inactivity (AOR = 2.17; 0.91–5.18), upper tercile of nutrient-dense foods (AOR = 1.35; 0.62–2.93), fat and protein diet (AOR = 1.31; 0.66–2.62), and cereal diet consumption (AOR = 3.87; 1.66–9.02).

## Conclusion

IBG and central obesity were prevalent and predicted by upper tercile consumption of nutrient-dense foods, high fat and protein diets, processed foods, and cereal diets, which could guide dietary interventions.

## 1. Introduction

Now a day the lifestyle characteristics of individuals, including dietary intake, change from time to time dramatically. At the same time, the occurrence of non-communicable diseases (NCDs) associated with impaired blood glucose (IBG) and obesity is rising [1]. Hence, these are early indicators of increased risk of NCDs and their complications [2, 3]. NCDs are responsible for the deaths of 41 million people (71%) each year, with low- and middle-income countries accounting for more than 85% of all deaths. Obesity and high blood glucose kill about 17.9 and 1.6 million people, respectively [4]. The burden of NCDs has significantly increased, mainly attributable to dietary habit change, increasing obesity and poor lifestyles as a result of poor lifestyle practices [5]. These are becoming major public health problems, especially in developing countries, accounting for an estimated 80% of all mortality [6].

Prediabetes or IBG, defined as fasting blood glucose (FBS) of 100–126 mg dl$^{-1}$, is a precursor to established diabetes mellitus, which could be aggravated by insulin resistance due to abdominal obesity [7]. In particular, central obesity, with a higher waist circumference (WC) and/or waist-to-hip circumference ratio (WHCR), is the sole cause of many NCDs related morbidity and mortality [8]. Among the risk factors, dietary consumption is a long-time exposure and one of the modifiable and preventable exposures for overt NCDs [3], which needs to be investigated systematically [9]. Dietary risk factors were responsible for an estimated 11 million (10–12) deaths and 255 million (234–274) DALYs lost; including high sodium intake, (3 million deaths), low intake of whole grains (3 million deaths) and low intake of fruits (2 million deaths) [9].

Due to the methodological shortcomings of individual food or nutrient analysis for predicting health outcomes, dietary pattern analysis is a better approach to characterize dietary consumption. It is becoming a more practical approach to determine the role of diet in the prevention and treatment of chronic diseases [2], taking into account the confounding, interaction, and synergistic effects of nutrients in food that make causation difficult [10, 11]. It helps to evaluate the cumulative and interactive effects of each dietary compound in a food and to consider the complexity of the diet [10, 12].

Hence, dietary patterns could be more predictive for IBG and central obesity. Central obesity and IBG are the major risk factors that increase the risk of NCDs, including diabetes mellitus. NCDs are a public health problem in Ethiopia, where 4.2% are victims of diabetes mellitus,

causing 70% of overall deaths [13]. While an estimated up to 39% are centrally obese, 5% of overall mortality is due to IBG [14]. In Ethiopia, an estimated 20.7% and 6.9% of adults are overweight and obese [15], where 35 to 38% of adults have central obesity [16]. This huge burden of central obesity, is one main risk factor for established NCDs and all their complications [17, 18]. For instance, an estimated 25% of IBG will progress to diabetes while 50% remain in an impaired state [19, 20] and increases cardiovascular mortality by 43% [21]. The rate of progression to visible diseases is rapid, necessitating a targeted intervention to slow progression.

Given the huge role of diet in the development of central obesity and IBG, there is a need to investigate the link in the absence of such evidence in Ethiopia. The study area is characterized by physical access to diversified diet from agricultural productivity, industrial processing and imports. Studies conducted in other countries showed a clear link between dietary patterns and the risk of central obesity and IBD [22–25]. However, evidence from other countries may not be applicable to our case, where dietary habits greatly vary even from region-to-region. In addition, studies conducted so far mainly focus on individual food consumptions [13, 16, 26, 27], which cannot be powerful predictors. This would help to design appropriate context-specific dietary strategies to decrease the risk of central obesity and IBG in Ethiopia. This study was to evaluate the association between major dietary patterns and central obesity and IBG among adults in Ethiopia.

## 2. Materials and methods

### 2.1. Study area and period

This study was conducted in Dire Dawa and Harar, located in the eastern parts of Ethiopia. Harar is located 526 kilometers from the capital city, Addis Ababa. Dire Dawa has an estimated population of more than 0.5 million, while more than 0.3 million people reside in the Harari regions, where more than half of the population resides in urban areas. According to the 2010 Harari Region population projection, there are 250, 093 with 146, 913 living in the urban areas and 122, 942 are males, with a total household of 64, 334. These two regions have a multiethnic and diverse population with a heterogenous agricultural productivity for fruits, vegetables, sorghum, maize, wheat, and other cereals. Furthermore, being an industrial center for many manufacturing industries and a trade center closest to neighboring countries makes access to imported foods easier, in addition to domestic production. Hence, rice, white flour, macaroni, and other industrially processed foods are accessible, in addition to cereals and grains marketed from other regions of the country. The study was conducted from September 10 to November 06, 2021. The study period was considered the fasting and feasting period to capture habitual Cereals, including teff, wheat, and legumes, were the most staple crops in the study areas. In addition, some processed foods, potatoes, and other fruits are the major foods consumed in the study area.

### 2.2. Study design and population

A community-based cross-sectional study was conducted among 501 randomly selected adults (aged > = 18 years) from the two study sites available during the study period. Hence, the findings of this study would be generalized to all adults residing in Harar and Dire Dawa towns, Eastern Ethiopia. Adults who were pregnant, had abdominal swelling (ascites), or had detectable thoracic and lumbar spinal deformity were excluded because waist circumference (WC) measurements are biased. Also, clients with serious psychiatric disorders, which make them unable to give oriented responses and unable to communicate, were excluded. More importantly, adults with clinically diagnosed diabetes and/or known diabetes patients on treatment were also excluded from the study.

## 2.3. Sample size determination and sampling procedures

The minimum sample size for this study was estimated using a sample size formula for single proportion with"P" as the prevalence of central obesity from the previous study at 24.4% [26], at a 95% confidence level, a "Z" critical value of 1.96, and a marginal error of "d" of 5%. The sample size became 145. The sample size for the association of dietary pattern with central obesity and IBG was calculated using the StatCalc module in Epi Info software version 7.0. After checking for many risk factors, the maximum sample size calculated was taken considering a prudent dietary pattern for predicting IBG (AOR = 2.26), abnormal fasting blood glucose level (OR = 2.26), percentage of IBG among those with a healthy dietary pattern (24%) [28], power of 80%, and equal sample of potential exposed and unexposed groups. Thus, taking dietary pattern as a risk factor for IBG and central obesity, the sample size became 504. After adding a non-response rate of 5%, the final minimum sample for this study was 526.

Stratified random sampling combined with simple and systematic random sampling was employed to select study participants. The total sample size was proportionally allocated to the two study sites (Dire Dawa and Harar) and to the smaller study units consecutively. A random sample of four districts in Harar and four kebeles (fourth administrative division level in Ethiopia) were randomly selected, where households were selected using systematic random sampling at a skipping interval. The sample interval (K) was calculated by dividing the total number of households by the allocated sample size for each kebele. Then, a random sample of adults residing in the selected household was selected and interviewed. The first household was selected randomly, and adults residing in randomly selected households were included in the study.

## 2.4. Data collection

Data were collected through a face-to-face interview using a set of pretested and structured questionnaires, as well as standardized anthropometric measurements and FBS measurement. The tool contains socio-demographic variables, lifestyle-related questions, and a validated semi-quantitative Food Frequency Questionnaire (FFQ) to capture the dietary intake of adults over the last one month. Data was collected by trained health care workers and graduating health students through face-to-face interviews, measurements, and laboratory analysis. Data collectors got the data by interviewing the study subject and probing for possible missed food consumed during the specific period of time.

The 16-item validated General Physical Activity Questionnaire (GPAQ) tool developed by WHO for physical activity surveillance was employed to assess the level of physical activity through face-to-face interviews. The activity level of the study participants was evaluated according to the standard WHO total physical activity calculation guide, and the level of total physical activity was categorized as physically active or inactive as detailed in the previous section. After calculating the metabolic equivalent per week (MET/week), a value below 600 was used to indicate physical inactivity [29].

A modified semi-quantitative Helen Keller FFQ containing 89 food items with about nine food groups was used to collect the dietary intake of study subjects. The FFQ was asked for the food that the subject consumed over the past 1 month to capture reliable and habitual dietary consumption of the individual. We further contextualized the list of food items for the particular context by considering the local foods commonly produced and consumed (S1 File). The FFQ validated for the Ethiopian adults has been contextualized and pretested to capture the dietary intake of adults. A validation study showed that the contextualized tool was valid and reliable in measuring the dietary micro- and macronutrient intake of adults in Ethiopia [30].

In pilot analysis, the FFQ tool we assed was found to be highly reliable with a Cronbach's alpha value above 0.86, indicative of a reproducible tool [31].

For assessment of central obesity, WC and hip circumference (HC) measurements were taken by trained anthropometric measurers using a non-stretchable tape meter at the end of a gentle expiration. The clothing and jewelry of the clients will be kept minimal. The WC was measured at a point midway between the lowest rib and the iliac crest in a horizontal plane at around the umbilicus while the respondents were instructed to breath gently. The tape meter is not tightly secured to avoid pressure and bias in measurement. While the HC was measured at the maximum circumference of the pelvis over the great trochanters, Measurements were made at least twice, and the average of the two measurements was recorded.

The laboratory analysis for blood glucose measurement was done by qualified and trained health professionals. The blood glucose measurement was done using a 2 mL capillary blood sample collected from the finger tips, and analyzed using a simple yet valid glucometer machine and recorded as mg dl$^{-1}$. A specific cut off point from 100 to 125 mg dl$^{-1}$ was used to diagnose IBG or prediabetes.

## 2.5. Data quality assurance

A pair of trained data collectors were deployed to collect the data from study subjects, as anthropometry needs curious measurements. Training was given on appropriate interview techniques and anthropometric measurements on WC and HC. Anthropometric reliability was assessed on 10 study subjects during the pretest, and intra- and interobserver variations were calculated. Data collectors with high inter and both intra-observer bias (technical error of measurement above 2%) were retrained and collected the data once it was in the acceptable range [32]. All standard measuring procedures and instruments were strictly followed during data collection. The standard operating procedure for blood sampling and glucose testing was followed accordingly. During data entry into EpiData, the data quality was maintained by making legal ranges, skipping patterns, appropriate coding, and careful data entry. A Piot study was conducted on the unselected study area in order to test and correct the necessary corrections. The FFQ is adapted and modified with the purpose of not missing relevant foods that are commonly consumed in the study area [33].

## 2.6. Data analysis

After checking for completeness and inconsistencies, the collected data was entered into Epi-Data Version 3.02 and exported to SPSS for analysis. All statistical analyses were carried out with Microsoft Excel 2016 and IBM SPSS Statistics for Windows, Version 20.0 (IBM Corp., Armonk, NY, USA). Descriptive statistics were computed for each variable, and the data was presented in statistical tables and graphs. The WHCR was calculated from WC and HC and checked for normality using the Kolmogorov-Smirnov test. Principal component analysis was conducted using household assets to derive the socioeconomic status (wealth index) of adults. All assumptions of PCA were checked using the Kaiser Mayer-Olkin (KMO) sampling adequacy, correlations, and covariance parameters. Based on the derived factor scores, wealth status was categorized into five quartiles, from the poorest to the wealthiest.

**Dietary pattern analysis.** Exploratory factor analysis using a principal component analysis was used to derive dietary patterns using the average frequency of 89-item food consumption in the past one month. The consumption frequency was recategorized into daily equivalents based on previous literature [34] and weighted daily consumption equivalents (**S2 File**). In addition, some closely related food items were summed up and recategorized into a single food item as needed. Bartlett's test of sphericity and the KMO measure of sampling

adequacy were used to verify the appropriateness of exploratory factor analysis. For the degree of intercorrelations between variables, we used a KMO value of greater than 0.5 and a p-value of less than 0.05 for the Bartlett test of sphericity. Orthogonally rotation using the varimax option was used to enhance the interpretability of dietary patterns [35]. The principal components were retained based on the following criteria: factor eigenvalue, identification of a break point in the scree plot, the proportion of variance explained, and interpretability of the dietary patterns [12, 36]. A rotated factor loading matrix was used to assess the direction and strength of correlation between each food item and its components. We also checked for a complex association between one food items and multiple factors, where such factors were removed. The corresponding factor score for each dietary pattern was constructed by summing the observed daily frequency of consumption of individual food items weighted by the factor loading. Hence, a high score of a component indicates more frequent consumption of the foods constituting that particular dietary pattern. For further analysis, each factor score was ranked into three terciles as low, medium, and high frequency of consumption.

**Dietary pattern with central obesity and IBG.** A separate multivariable logistic regression model was fitted to evaluate the association between dietary pattern and central obesity and IBG. Both crude and adjusted odds ratios with 95% confidence intervals and/or p-values were computed. Multicollinearity was checked using the variance inflation factor (VIF) and/or the standard error estimates. The model fitness was checked using Hosmer and Lemeshow's goodness of fit. In addition, variables with a p-value below 0.2 in bivariable logistic regression and biologically plausible risk factors for the outcomes were considered in the multivariable logistic models. Statistical significance was declared at a p-value below 0.05.

## 2.7. Variables of the study

While the dependent variables of this study were IBG and central obesity, the independent variables were sociodemographic variables (age, sex, income, occupation, and education), physical activity, family history of diabetes, dietary pattern, dietary intake of food groups, and habits of substance use. The outcome variable IBG was categorized as IBG when the FBS between 100 and 125 mg dl$^{-1}$ [37]. Central obesity was defined as having a waist circumference of 94 cm for men and 80 cm for women, or a WHCR of 0.90 in men and 0.85 in women [38].

## 2.8. Ethical approval

Ethical approval was obtained from Dire Dawa University, Institutional Research Review Board. The support letter was taken to the respective regional health office for cooperation. A written informed consent was obtained from each study participant after explaining the study purpose, confidentiality, and procedures in detail. The study was conducted in accordance with the Declaration of Helsinki for Human Research [39]. The data will be used for this research purpose only, and the data will be kept confidential. Respondents identified with central obesity and IBG got counseling, recommendation, and/or linkage to the health facilities. All the COVID-19 standard prevention precautions were implemented throughout the data collection process.

## 3. Results

### 3.1. Sociodemographic characteristics

A total of 501 study subjects were included in the current survey, with a response rate of 95.3%. More than half (66.7%) of the participants were females. The age of the respondents ranges from 18–64 years, with a mean age of 41 years (±12). The majority, 90% of the

respondents reside with their families. A total of 203 (40.5%) attended primary school, and 143 (28.5%) and 127 (25.3%) were housewives and government employees, respectively. About 188 (37.5%) and 44 (8.8%) were in the wealthier and wealthiest ranks regarding socioeconomic status. On the other hand, 105 (21%) and 100 (20%) of the respondents were from the poorest and poorer households, respectively (Table 1).

### 3.2. Physical activity and substance use related factors

Regarding the physical activity of adults, more than three-fourths (383, 76.8%) of them were reported to be physically inactive with a median metabolic equivalent of 240 MET (IQR: 0–500 MET). While only 55 (11%) had a regular physical activity schedule, 85 (27.9%) had a family history of diabetes mellitus. Approximately 304 (60.7%) reported chewing khat, 101 (20.2%) reported drinking alcohol, and 24 (4.8%) reported chewing khat and smoking cigarettes.

### 3.3. Dietary patterns of adults

A total of 465 (98%) of adults had a usual three-times-a-day meal, while 478 (98.4%) were reported to have a regular meal. Almost all (99.4%) reported using liquid oils for cooking. About 95 (19%) and 48 (9.6%) reported having regular sweet and soda drinks and fatty foods, respectively. In terms of fruit and vegetable consumption, only 96 (19.2%) reported regular fruit consumption and 140 (27.9%) reported vegetable consumption (at least $> = 6$ times per week).

The consumption frequency was recategorized into daily equivalents based on previous literature and weighted daily consumption equivalents. In addition, some closely related food items were summed up and categorized into a single food item as needed. An exploratory factor analysis using the principal component analysis method was conducted. Food items with a KMO value of below 0.30 were removed iteratively. All the assumptions of PCA were checked. The overall sample adequacy was evaluated using KMO, where the p-value was 0.91. Similarly, the presence of interitem correlation was checked using the Bartlett test of sphericity (p-value 0.0001), indicating the presence of a statistically significant correlation. Furthermore, items with a communality below 0.30 were removed step-by-step. The final food items to be included were determined by the following criteria: interpretability of the factors, improving the total variance explained, absence of complex structure evidenced by rotation component matrix, eigen value, and scree plots (S3 File).

Five dietary patterns explaining 71% of the total variance were identified; "Animal source foods, fruits, and vegetables-nutrient dense foods", "high fat and protein foods", "processed foods", "traditional alcoholic drinks", and "cereal foods". A total of sixteen, nine, seven, four and four food items constitute the first, second, third, fourth, and fifth dietary patterns as indicted in S3 File. Nutrient dense foods composed of animal source foods, fruits, and vegetables account for the major part of the variance (42%), followed by high fat and protein foods (11.2%). For further analysis, the standard score of each dietary pattern was categorized into three terciles, where the lower score indicates less frequent consumption of foods and is assigned the lowest tercile. Furthermore, traditional alcoholic drinks were retained as they are relevant to central obesity and IBG (Table 2).

### 3.4. Magnitude of central obesity and impaired blood glucose

As clearly shown in Fig 1 below, an estimated 20.4% (95% CI: 17.0–24.2%) of the subjects had impaired blood glucose or prediabetes. While 14.6% (95% CI: 11.8–17.9%) of adults had central obesity based on the sex-specific WC cutoff points, 94.6 (95% CI: 92.3–96.3) of them had a WHCR, which makes them at risk for any cardiovascular complications (Fig 1). When

**Table 1. Sociodemographic characteristics of respondents for a dietary pattern study in Dire Dawa and Harar, eastern Ethiopia.**

| Variables | Categories | Frequency | Percent |
|---|---|---|---|
| Sex | Male | 182 | 33.3 |
| | Female | 319 | 66.7 |
| Live with | With family | 450 | 90 |
| | Friends | 28 | 5.6 |
| | Lonely | 21 | 4.4 |
| Education | Illiterate | 24 | 4.6 |
| | Primary education | 203 | 40.5 |
| | Secondary education | 107 | 21.4 |
| | College and above | 167 | 33.3 |
| Marital status | Married | 409 | 81.6 |
| | Single | 71 | 14.2 |
| | Widowed | 21 | 4.2 |
| Occupation | Daily laborer | 18 | 3.6 |
| | Farmer | 9 | 1.8 |
| | Government employee | 127 | 25.3 |
| | Housewife | 143 | 28.5 |
| | Merchant | 69 | 13.8 |
| | Private employed | 61 | 12.2 |
| | Others | 74 | 14.8 |
| Wealth index | Poorest | 105 | 21.0 |
| | Poor | 100 | 20.0 |
| | Medium | 64 | 12.8 |
| | Wealthier | 188 | 37.5 |
| | Wealthiest | 44 | 8.8 |

disaggregated by sex, females had a higher rate of central obesity by WHCR (98% vs. 86%) and WC (17.2% vs. 9.9%). Moreover, the prevalence of IBG was higher among males (22.5%) compared to females (17.5%). The mean blood glucose level was 180.1 mg dl$^{-1}$ (±50). Similarly, the mean and standard deviation for WC and WHCR were 66.7 (±13.4) and 0.96 (±0.46).

## 3.5. Factors associated with central obesity and impaired blood glucose

An exploratory logistic regression was carried out to identify the association between sociodemographic factors and dietary patterns with IBG and central obesity among adults, as detailed in Table 3. Increasing age is associated with an increased odds of central obesity and IBG. While males (COR = 1.24; 95% CI: 0.79–1.93) had 24% higher odds of IBG compared to females, females (COR = 1.86; 1.05–3.30) had a significantly higher occurrence of central

**Table 2. Total variance explained and summary statistics for factor scores for identified dietary patterns in Ethiopia.**

| S.no | Dietary patterns | Label | Variance explained | Median (Q$_2$) | Q$_1$ | Q$_3$ |
|---|---|---|---|---|---|---|
| 1 | DP_1 | Animal source foods, fruits, and vegetables-Nutrient dense foods | 42% | -0.250 | -0.358 | -0.180 |
| 2 | DP_2 | High fat and protein foods | 11.2% | -0.036 | -0.241 | 0.079 |
| 3 | DP_3 | Processed foods | 9.0% | -0.460 | -0.762 | 0.667 |
| 4 | DP_4 | Traditional alcoholic drinks | 5.0% | -0.146 | -0.176 | -0.056 |
| 5 | DP_5 | cereal foods | 3.83% | -0.177 | -0.265 | 0.087 |
| | Total | | 71.0% | - | - | - |

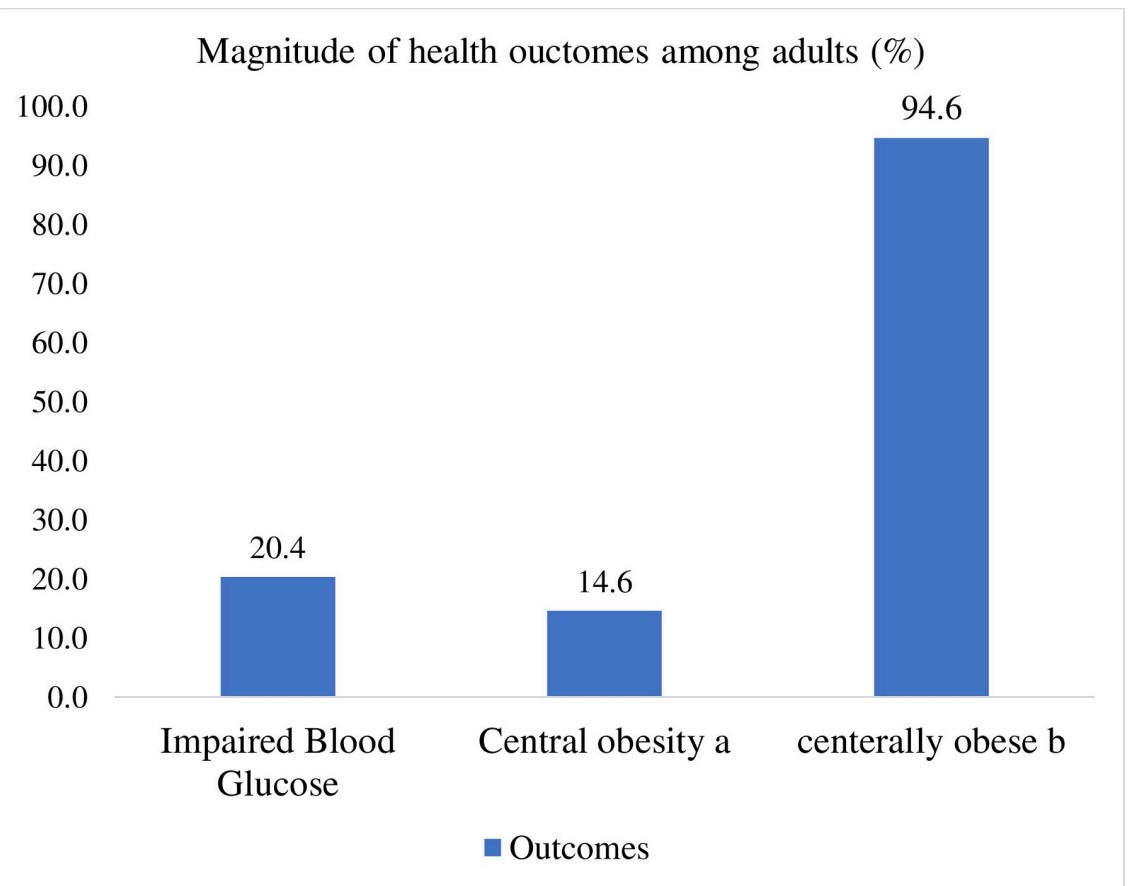

**Fig 1.** The magnitude of impaired blood glucose, central obesity (based on WC cutoff points of 80 and 94 cm for females and males, [a] WHC of 0.85 and 0.90 for females and males, [b]) among adults residing in Dire Dawa and Harar, Eastern Ethiopia.

obesity. The likelihood of IBG (COR = 2.18; 95% CI: 1.34–3.53) and central obesity (COR = 9.65; 95% CI: 4.29–21.7) was 2.1 and 9.65-times higher in the wealthiest versus poorest families, respectively. Similarly, housewives had 38% increased odds of IBG (COR = 1.38; 95% CI: 0.84–2.28) and central obesity (COR = 2.33; 95% CI: 1.34–4.04) compared to adults engaged in private work. The likelihood of central obesity was found to be high among those with irregular meals (COR = 1.68; 95% CI: 0.60–4.66), physically inactive (COR = 1.54; 95% CI: 0.95–2.503), and habits of sweet food consumption (COR = 1.63; 95% CI: 0.91–2.91) (Table 3).

Dietary patterns were found to be associated with higher odds of central obesity and IBG among adults. Adults who consumed less nutrient dense food (COR = 1.76; 95% CI: 1.05–2.95) and more fat and protein-rich foods (COR = 1.35; 95% CI: 0.77–2.35), for example, had a 1.76- and 1.35–times higher likelihood of having IBG, respectively. More frequent consumption of processed foods (COR = 3.75; 95% CI: 2.08–6.76) and medium consumption of traditional alcoholic drinks (COR = 1.37; 95% CI: 0.82–2.28) were significant predictors of IBD. However, optimal consumption of a cereal diet was protective against IBG. Furthermore, eating nutrient-dense foods (COR = 1.50; 95% CI: 0.87–2.59) and highly processed foods (COR = 1.74; 95% CI: 0.95–3.20) showed to increase the odds of central obesity by 50% and 74%, respectively. Frequent consumption of traditional alcoholic drinks (COR = 1.57; 95% CI: 0.89–2.76) and consumption of cereal diets (COR = 3.98 (2.05–7.72)) was shown to increase

**Table 3. Bivariable logistic regression output for determinants of IBG and central obesity among adults from Dire Dawa and Harar, Eastern Ethiopia.**

| Factors | Categories | IBG (pre diabetes) | | COR (95% CI) | Central obesity (WC) | | COR (95% CI) |
|---|---|---|---|---|---|---|---|
| | | IBG | No IBG | | Yes | No | |
| Age in years | | – | – | 1.01 (0.99–1.03) | – | – | 1.03 (1.01–1.1) |
| Sex | Female | 61 | 258 | 1 | 55 | 264 | 1.86 (1.05–3.30) * |
| | Male | 41 | 141 | 1.24(0.79–1.93) | 18 | 164 | 1 |
| Wealth index | Poor | 30 | 175 | 1 | 7 | 198 | 1 |
| | Medium | 9 | 55 | 0.96 (0.43–2.13) | 7 | 57 | 3.47 (1.17–10.3) * |
| | Wealthier | 63 | 169 | 2.18 (1.34–3.53) * | 59 | 173 | 9.65 (4.29–21.7) * |
| Occupation | Housewife | 35 | 106 | 1.38 (.84–2.28) | 34 | 107 | 2.33 (1.34–4.04) * |
| | Government | 22 | 105 | 0.88 (.50–1.54) | 11 | 116 | 0.69 (0.33–1.45) |
| | Private | 45 | 188 | | 28 | 205 | 1 |
| Regular meals | No | 1 | 23 | 1 | 5 | 18 | 1.68 (0.60–4.66) |
| | Yes | 102 | 375 | | 68 | 410 | 1 |
| Physical activity | Physically active | 31 | 88 | 1.54 (0.95–2.50) | 1 | 118 | 1 |
| | Physically inactive | 71 | 311 | 1 | 72 | 310 | 1.54 (0.95–2.503) |
| Sweet and soda drinks | No | 75 | 331 | 1 | 54 | 352 | 1 |
| | Yes | 27 | 68 | 1.75 (1.05–2.92) * | 19 | 76 | 1.63 (0.91–2.91) |
| Regular physical activity | No | 94 | 352 | 1.57 (0.72–3.43) | 64 | 382 | 0.86 (0.40–1.83) |
| | Yes | 8 | 47 | 1 | 9 | 46 | |
| Nutrient dense foods (DP-1) | Lowest | 45 | 119 | 1.76 (1.05–2.95) * | 27 | 137 | 1 |
| | Middle | 26 | 136 | 0.89 (0.50–1.57) | 6 | 156 | 0.20 (0.08–0.49) |
| | Highest | 31 | 144 | 1 | 40 | 135 | 1.50 (0.87–2.59) |
| High fat and protein diet (DP-2) | Lowest | 41 | 126 | 1.70 (1.0–2.92) | 33 | 134 | 1.63 (0.91–2.94) |
| | Middle | 34 | 132 | 1.35 (0.77–2.35) | 18 | 148 | 0.81 (0.42–1.57) |
| | Highest | 27 | 141 | 1 | 22 | 146 | 1 |
| Processed foods (DP-3) | Lowest | 18 | 148 | 1 | 20 | 146 | 1 |
| | Middle | 32 | 137 | 1.92 (1.03–3.58) * | 21 | 148 | 1.04 (0.54–1.99) |
| | Highest | 52 | 114 | 3.75 (2.08–6.76) * | 32 | 134 | 1.74 (0.95–3.20) |
| Alcoholic drinks (DP-4) | Lowest | 34 | 133 | 1 | 24 | 142 | 1 |
| | Middle | 31 | 132 | 1.37 (0.82–2.28) | 13 | 150 | .51 (0.25–1.05) * |
| | Highest | 37 | 134 | 0.68 (0.38–1.20) | 36 | 136 | 1.57 (0.89–2.76) |
| Cereal diets (DP-5) | Lowest | 34 | 133 | 1 | 13 | 154 | 1 |
| | Middle | 31 | 132 | .092 (0.53–1.58) | 17 | 146 | 1.38 (0.65–2.94) |
| | Highest | 37 | 134 | 1.08 (0.64–1.82) | 43 | 128 | 3.98 (2.05–7.72) * |

Note: * statistically significant association at p-value <0.05.

the occurrence of central obesity by 57% and 258%. However, in a dietary pattern characterized by fat and protein-rich foods, the risk for both central obesity (COR = 1.70; 95% CI: 1.0–2.92) and IBG (COR = 1.63; 95% CI: 0.91–2.94) was higher among those with a lower tercile (Table 3).

A separate multivariable logistic model was fitted in a step-wise backward regression approach for central obesity and IBG. Sex, wealth status, physical activity level, and dietary patterns 1, 3, and 5 were found to be associated with central obesity among adults. Being physically inactive (600 MET) (AOR = 21.1; 95% CI: 2.77–161.4), from a higher socioeconomic class (6.92 (2.91–16.5), and females (AOR = 1.67 (0.82–3.40)) were positively associated with a higher occurrence of central obesity. More importantly, those who consumed more nutrient-

dense foods (1.75; 95% CI: 0.75–4.06), processed foods (AOR = 1.41 (0.57–3.48), and cereal diets (AOR = 4.06; 95% CI: 1.87–8.82) had a higher likelihood of having central obesity. An increasing dose-response relationship was observed where odds of its occurrence was higher for more frequent consumption (Table 4).

A multivariable logistic regression output was fitted using Hosmer and Lemeshow's goodness of fit (P-value = 0.847), indicating a fit model. Those from wealthier families (AOR = 2.36; 95% CI: 1.36–4.10), those without regular physical activity (AOR = 2.17; 95% CI: 0.91–5.18) and those with sweet food consumption habits (AOR = 2.46; 95% CI: 1.30–4.66) had a 2.36-, 2.17-, and 2.46-times increased the likelihood of having IBG level compared to counterparts. Among the identified dietary patterns, lower consumption of nutrient-dense foods (AOR = 1.35; 0.62–2.93), higher fat and protein diets (AOR = 1.31; 95% CI: 0.66–2.62) and higher cereal diet consumption (AOR = 3.87; 95% CI: 1.66–9.02) were important dietary determinants of IBG level. On the other hand, the odds of IBG were 60% higher among those who were physically active (AOR = 1.60; 95% CI: 0.90–2.85) (Table 5).

## 4. Discussion

The burden of NCDs is rising at an alarming rate, accounting for more than 70% of adult mortality [40]. Prediabetes and central obesity associated with metabolic syndrome are the most important risk factors for the development of overt NCDs [41]. These diseases are in turn dependent on the lifestyle and dietary consumption habits of individuals, which could be modified for a better health condition [42]. Hence, we conducted the current study to identify the role of dietary patterns in the development of prediabetes (IBG) and central obesity among adults.

In this study, we found that 20.4% had IBG, 14.6% had central obesity, and 94% had an increased risk for cardiovascular diseases based on the sex-specific WHCR cutoff point. This indicates IBG is a common public health problem, especially among females, and it may be cross linked with a comparable occurrence of central obesity (abdominal obesity) based on WC cutoff points. Evidence from Ethiopia also showed that the prevalence of IBG ranges from 8% [43] to 12% [44]. It should be emphasized that central obesity is a main risk factor for insulin resistance and the development of IBG. Since 85% of NCD deaths occur in low- and middle income countries [4], multilateral dietary and lifestyle interventions are required to halt central obesity and IBG. This will be needed to decrease the incidence of metabolic risk factors before an overt metabolic disease (cardiovascular or diabetes) happens. A higher prevalence of central obesity (39%) was reported in southern Ethiopia [45].

In the present study, being female, higher socioeconomic class (AOR = 6.92; 95% CI: 2.91–16.5), and physical inactivity (AOR = 21.1; 95% CI: 2.77–161.4) were found to be important factors associated with central obesity. A similar association was also identified with IBG, except that being physically active is associated with a higher odd of IBG. This was also supported by findings from Ethiopia, where females had a 9-times likelihood (AOR = 9.62; 95% CI: 4.84–19.12) [16] and 12-times likelihood (AOR = 12.93, 95% CI: 6.74–24.79) for having higher prevalence of central obesity compared to males. A similar association was also reported for socioeconomic class [46], where poor lifestyles and access to and consumption of unhealthy foods are more common. In addition, physical activity and having regular physical activity had a crucial for maintaining energy balance. A study indicated that adults who were physically active were less likely to have central obesity (AOR = 0.87: p-value 0.174), emphasizing the need for regular physical activity [46].

On the other hand, the role of non-exercise energy expenditures like working, walking, and others should be encouraged as these are very crucial for maintaining optimal fat storage [47,

**Table 4. Adjusted association between dietary patterns and central obesity among adults from Dire Dawa and Harar, Eastern Ethiopia.**

| Factors | Categories | Central obesity (WC) | | AOR (95% CI) | p-value |
|---|---|---|---|---|---|
| | | Yes | No | | |
| Sex | Female | 55 | 264 | 1.67(0.823.40) | 0.157 |
| | Male | 18 | 164 | 1 | |
| Wealth index | Poor | 7 | 198 | 1 | |
| | Medium | 7 | 57 | 2.56(0.80–8.25* | 0.115 |
| | Wealthier | 59 | 173 | 6.92(2.91–16.5) | 0.001** |
| Physical activity | Physically active | 1 | 118 | 1 | |
| | Physically inactive | 72 | 310 | 21.1(2.77–161.4) | 0.003** |
| Nutrient dense foods (DP-1) | Lowest | 27 | 137 | 1 | |
| | Middle | 6 | 156 | 0.28(0.09–0.87) | 0.028* |
| | Highest | 40 | 135 | 1.75(0.75–4.06) | .193 |
| Processed foods (DP-3) | Lowest | 20 | 146 | 1 | |
| | Middle | 21 | 148 | 1.59(0.73–3.46) | 0.246 |
| | Highest | 32 | 134 | 1.41(0.57–3.48) | 0.456 |
| Cereal diets (DP-5) | Lowest | 13 | 154 | 1 | |
| | Middle | 17 | 146 | 1.51(0.66–3.45) | 0.325 |
| | Highest | 43 | 128 | 4.06(1.87–8.82)* | 0.0001** |

Note: statistically significant factors associated with central obesity at p-value <0.05* and 0.001**

48]. However, the negative association with IBG could be explained by the fact that those with some risk of cardiovascular complications might have poor blood sugar levels. This does not

**Table 5. Adjusted multivariable logistic regression showing factors associated with IBG among adults residing in Dire Dawa and Harar, Eastern Ethiopia.**

| Factors | Categories | IBG (pre diabetes) | | AOR (95% CI) | P-value |
|---|---|---|---|---|---|
| | | IBG | No IBG | | |
| Wealth index | Poor | 30 | 175 | 1 | |
| | Medium | 9 | 55 | 1.29(0.54–3.04) | 0.567 |
| | Wealthier | 63 | 169 | 2.36(1.36–4.10) | 0.002* |
| Physical activity | Physically active | 31 | 88 | 1.60(0.90–2.85) | 0.107 |
| | Physically inactive | 71 | 311 | 1 | |
| Sweet and soda drinks | No | 75 | 331 | 1 | |
| | Yes | 27 | 68 | 2.46(1.30–4.66) | 0.006* |
| Regular physical activity | No | 94 | 352 | 2.17(0.91–5.18) | 0.083 |
| | Yes | 8 | 47 | 1 | |
| Nutrient dense foods (DP-1) | Lowest | 45 | 119 | 1.35(0.62–2.93) | 0.490 |
| | Middle | 26 | 136 | 1.27(0.65–2.5) | |
| | Highest | 31 | 144 | 1 | |
| High fat and protein diet (DP-2) | Lowest | 41 | 126 | 1.42(0.75–2.71) | 0.285 |
| | Middle | 34 | 132 | 1.31(0.66–2.62) | 0.443* |
| | Highest | 27 | 141 | 1 | |
| Processed foods (DP-3) | Lowest | 18 | 148 | 1 | |
| | Middle | 32 | 137 | 1.97(1.01–3.86) | 0.048* |
| | Highest | 52 | 114 | 3.87(1.66–9.02) | 0.002** |

Note: statistically significant factors associated with central obesity at p-value <0.05* and 0.001**

invalidate the positive role of physical activity for optimal weight and blood glucose. Rather, it reflects the characteristics of the study subjects.

Although there is evidence for the link between some dietary consumption habits and central obesity and IBG [49], more concrete and context-specific evidence using a more robust approach is needed to guide obesity and diabetes prevention strategies [50]. Due to the complex and multicollinear nature of the human diet and the limitations of individual food or nutrient analysis for predicting a particular health outcome, dietary pattern analysis using a more robust statistical approach is becoming a novel approach for better recommendations [12, 36]. Hence, we derived the dietary patterns and its association with the development of central obesity and IBG.

Those who reported consuming processed foods (AOR = 1.41; 95% CI: 0.57–3.48) and cereal diets (AOR = 4.06; 95% CI: 1.87–8.82) were major factors associated with the development of central obesity. Other studies indicated that the top tercile of western pattern consumption (OR = 1.82; 95% CI: 1.16–2.87) increased the likelihood for obesity while a prudent pattern (AOR = 0.62; 95% CI: 0.40–0.96) reduced their likelihood for central obesity [51]. On the other hand, those with a low consumption of nutrient dense foods (animal sources, fruits, and vegetables) (AOR = 1.35; 95% CI: 0.62–2.93) had a 35% increased risk of IBG compared to those with a higher tercile. This could be explained by undernourished adults' metabolic limitations and an increased risk of insulin insufficiency for glucose metabolism [52].

Other studies also showed that dietary patterns were highly correlated with abnormal blood glucose status (r = 0.8) [53]. A review of the evidence also found that healthy, traditional, and Mediterranean diets rich in sea foods, fruits, and vegetables were effective in preventing IBG [54]. Similarly, higher adherence to the unhealthy dietary pattern (saturated fats and meat) was more likely to cause insulin resistance and IBG as a result [49]. On the contrary, a vegetarian dietary pattern reduced the occurrence of IBG (p-value below 0.05) [28]. For instance, a study showed that a higher tercile of the western dietary pattern was associated with an increased likelihood of having central obesity (AOR = 1.80; 95% CI: 1.15–2.81) compared to a lower tercile [55].

The association between lower tercile consumption of nutrient-dense foods with IBG could be related to the issue of food security and the risk of having unhealthy dietary patterns. For example, a study found that food insecurity could lead to unhealthy dietary patterns and ultimately increase the likelihood of having central obesity [56]. The current study depicted the role of dietary pattern in the development of central obesity and prediabetes. These could help to design and indicate an effective dietary strategy to prevent obesity and its associated consequences [57].

## Strength and limitations of the study

The current study is novel and would fill existing knowledge gaps regarding concrete evidence on the relationship between dietary pattern and central obesity and IBG. Furthermore, we employed a more reliable dietary assessment approach and dietary pattern analysis using a more appropriate and feasible analytic strategy. However, the findings of this study should be in the light of some Methodological limitations. First, in assessing food consumption, the potential for recall bias and respondent bias could not be excluded, which might affect bias the overall dietary consumptions. Moreover, the current study did not collect quantitative data on daily intake to quantify the nutrient intake beyond food consumption. The current study mainly focuses on the urban setting, where the disease burden could be higher. However, a comparative study including urban and rural setting could better show the association in a very informative way.

## 5. Conclusion and recommendations

Overall, the occurrence of central obesity, IBG, and the risk for cardiovascular diseases were prevalent in the study area. Dietary patterns characterizing the overall diet of individuals are more predictive of these outcomes, which could be replicated for other relevant diet- disease studies to inform policy and strategies targeting the risks for non-communicable disease. Higher tercile consumption of nutrient dense foods, high fat and protein diets, processed foods, and cereal diets are important risk factors for central obesity and IBG. Optimal consumption of a diversified diet from animal sources, fruits, vegetables, cereals, and protein-rich foods could play a great role in preventing the occurrence of metabolic syndromes, while consumption of processed foods and alcoholic drinks should be kept to a minimum. The findings of this study will inform the existing policies and programs targeting non-communicable disease prevention in Ethiopia. We strongly recommend regional health bureaus, non-governmental organizations working on nutrition, and other concerned bodies to strategically implement appropriate behavioral change communication and agricultural strategies healthy diet for the wider community.

## Supporting information

**S1 File. Semi-quantitative food frequency questionnaire.**
(DOCX)

**S2 File. Conversion factors to a daily frequency of food consumption for adults.**
(DOCX)

**S3 File. Identified major dietary patterns (components) and their factor loading for individual food item among residing in Dire Dawa and Harar, Eastern Ethiopia.**
(DOCX)

## Acknowledgments

We are grateful to Dire Dawa University and Dire Dawa Health Bureau for their collaborations and support for the successful completion of the study. Our gratitude also goes to the respective health bureaus, respondents, data collectors, and supervisors for their sincere help and collaboration for the successful completion of the research.

**Ethical approval and consent to participate**

Ethical approval was obtained from the Institutional Research Ethical Review Board of Dire Dawa University. Written informed assent was obtained from all adults. This research was conducted in accordance with the Helsinki Declaration for protecting human study subjects.

## Author Contributions

**Conceptualization:** Berhanu Abebaw Mekonnen, Abdu Oumer, Ahmed Ale, Aragaw Hamza, Imam Dagne, Abdurezak Adem Umer, Dilnessa Fentie, Muluken Yigezu, Zerihun Tariku, Shambel Abate.

**Data curation:** Berhanu Abebaw Mekonnen, Abdu Oumer, Muluken Yigezu.

**Formal analysis:** Berhanu Abebaw Mekonnen, Abdu Oumer.

**Funding acquisition:** Abdu Oumer, Ahmed Ale, Aragaw Hamza.

**Investigation:** Abdu Oumer, Imam Dagne, Abdurezak Adem Umer, Dilnessa Fentie, Muluken Yigezu, Zerihun Tariku.

**Methodology:** Abdu Oumer, Imam Dagne, Abdurezak Adem Umer, Dilnessa Fentie, Zerihun Tariku, Shambel Abate.

**Project administration:** Abdu Oumer, Ahmed Ale.

**Resources:** Abdu Oumer, Ahmed Ale, Aragaw Hamza, Muluken Yigezu.

**Software:** Berhanu Abebaw Mekonnen, Abdu Oumer.

**Supervision:** Abdu Oumer, Ahmed Ale, Aragaw Hamza, Imam Dagne, Abdurezak Adem Umer, Dilnessa Fentie, Muluken Yigezu, Zerihun Tariku, Shambel Abate.

**Validation:** Berhanu Abebaw Mekonnen, Abdu Oumer.

**Visualization:** Abdu Oumer, Ahmed Ale, Abdurezak Adem Umer, Muluken Yigezu, Zerihun Tariku.

**Writing – original draft:** Abdu Oumer.

**Writing – review & editing:** Berhanu Abebaw Mekonnen, Abdu Oumer, Ahmed Ale, Aragaw Hamza, Imam Dagne, Abdurezak Adem Umer, Dilnessa Fentie, Muluken Yigezu, Zerihun Tariku, Shambel Abate.

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
