## [Decision Letter · Decision Letter 0]

2 Feb 2023

PONE-D-22-28499Major dietary patterns of community dwelling adults and their associations with impaired blood glucose and central obesity in Eastern Ethiopia: diet-disease exploration studyPLOS ONE

Dear Dr. Oumer,

Thank you for submitting your manuscript to PLOS ONE. After careful consideration, we feel that it has merit but does not fully meet PLOS ONE’s publication criteria as it currently stands. Therefore, we invite you to submit a revised version of the manuscript that addresses the points raised during the review process.

We look forward to receiving your revised manuscript.

Kind regards,

Mohammad Asghari Jafarabadi

Academic Editor

PLOS ONE

Journal Requirements:

Reviewers' comments:

Reviewer's Responses to Questions

**Comments to the Author**

1. Is the manuscript technically sound, and do the data support the conclusions?

Reviewer #1: Yes

Reviewer #2: Yes

2. Has the statistical analysis been performed appropriately and rigorously? 

Reviewer #1: Yes

Reviewer #2: Yes

3. Have the authors made all data underlying the findings in their manuscript fully available?

Reviewer #1: Yes

Reviewer #2: Yes

4. Is the manuscript presented in an intelligible fashion and written in standard English?

Reviewer #1: Yes

Reviewer #2: Yes

5. Review Comments to the Author

Reviewer #1: As identified in the manuscript, non-communicable diseases associated with impaired blood glucose and obesity are a public health burden, especially in developing countries. The manuscript provided appropriate background on why the study was conducted, provided significant detail on the research methodology, and used appropriate and validated research instruments.

The paper could be strengthened by more clearly delineating throughout the Discussion which quoted statistics/cited references are related to Ethiopia, which are specific to Sub-Saharan Africa, and which may be related to other countries/regions in the world. Such distinctions were made sometimes, but not consistently throughout the Discussion, which sometimes made the Discussion difficult to follow. For example, in the opening paragraph of the Discussion (p. 23) the 1st sentence is specific to Ethiopia, the 2nd sentence seems to reflect a more global view, and the 3rd sentence is based on a scoping review in Sub-Saharan Africa. In addition, suggest more fully describing in the Discussion the specific strengths and weaknesses of the current study.

In the Conclusion and Recommendations suggest editing the section to more clearly identify those statements which are specifically based on results of the study. For example, in the 1st sentence, suggest adding a phrase such as “in this study among randomly selected adults in Eastern Ethiopia” after the word “Overall” and then deleting “among adults” at the end of that sentence. Such edits will help to differentiate statements about the study vs other statements that may be more broad in nature (such as what the last statement in the paragraph appears to be). Also, in the Conclusions and Recommendations section, suggest adding further explanation of how the results link back to one of the stated purposes of the study, which was “to design appropriate context-specific dietary strategies to decrease the risk of central obesity and IBG in Ethiopia.”

The paper would benefit from minor English editing to confirm use of appropriate verb tenses and word choice as well as correct any minor spelling errors.

Reviewer #2: Pertinent questions, suggestions, and comments:

The research consists of noble ideas with great findings that will fill a research gap, particularly in Ethiopia. But, there are a significant number of issues that must be to be addressed by the authors. I mention the issues as follows.

1. The title

In the study's title, you wrote as it is an exploration type. Can we consider quantitative study as explorative? I know the explorative type of study are qualitative.

2. Methodological issues.

Which particular study design did the authors use? For instance, a cross-section, Cohort, or other. You mentioned “A community-based survey” in the section on methodology. This is not a study design. It is a data collection method. You must state your study design and why you select the specific design. Page 11, section 2.2. Study design and population

3. Result section: Page 17 to 23

Numerous areas need revision. I commented on the track change. You can find it in the attached document. However, I was supposed to ask critical questions. You stated a variable that is not in the section. For example, “(AOR = 1.60; 95% CI: 0.90–2.85) Page 23 line 6” From where you brought this. Table five did not have such findings. This is a serious issue. Another critical issue is the way you interpreted the ODDS result. ODDS are not just numbers. They are the likelihood or the chance of having certain characteristics. You did not do the risk ratio. As a result of this, you cannot interpret it as a risk of obesity or IBG.

4. Section Discussion page 23 to 25

This section, especially the association between dietary patterns and central obesity and IBG in this section, is not adequate. In addition, it is not exhaustive. You need to compare with similar findings and identify similar and unique variables. If this study identifies a unique variable, you need to have a possible justification.

5. Limitation and strength of the study section, page 25 lines 29 to 30

In this section, you need to state the strength and limitations of the study without hesitation. Additionally, how did you try to minimize recall bias? Because it will affect the findings of the dietary pattern.

6. Conclusion and Recommendation, page 26 lines 3 to 11

The recommendation is not plausible. As a result, it lacks a strategy to prevent the problem. Moreover, it was not indicated the specific responsible organ of the entity that can tackle the issue.

7. Section study area and period, page 11 line 8

You mentioned the study period from September to November. It is not correct. Specific dates must be mentioned. Because the dates may be from 30 September to 01 November 2021 or any other dates. It is extremely difficult to know the exact date of the study.

8. There are editorial problems in the whole document. You need to revise the entire document. I make the areas that need to be revised by track change.

9. There are a lot of comments those I am not able to mention here, So. I strongly advise the authors to check the attached documents.

6. PLOS authors have the option to publish the peer review history of their article (what does this mean?). If published, this will include your full peer review and any attached files.

Reviewer #1: No

Reviewer #2: No

---

## [Author Response · Author response to Decision Letter 0]

3 Feb 2023

PONE-D-22-28499

Major dietary patterns of community dwelling adults and their associations with impaired blood glucose and central obesity in Eastern Ethiopia: diet-disease exploration study

Dear editor, please kindly note that we have added authors which have significant contribution to the current work. We have attached the PLOS One authorship change request form filled for your approval.

Dear editor and reviewers

Thank you very much for having such detail and valuable comments on our work. We have checked your comments and suggestion in more detail and have included them in the revised submission. All the changes are showed in track changes. Hence, we found it that giving a detail response for each point is not necessary. Thus, we only responded to question which need clarification ad included the requested changes in the revised manuscript simply. 

On the title, the comment was reasonable and we amended the title accordingly. We also amended the comments on the abstract as it refers to percent based on WHCR cutoff point. These finding is discussed in more detail in the discussion part.

We specified the study design as survey or cross-sectional study design and the study period as “September 10 to November 06”.

You declared the final sample size as 526 in the section of sample size determination. Why you did you decide on 501?

The estimated minimum sample size required for this study was 526. However, the achieved response rate was 95% where a total 25 respondents were non-responders. They were not volunteer to participate in the study. The difference is due to this.

A separate multivariable model was done for IBG and central obesity as this is shown in Table 4 and 5. This is also clearly indicated there. 

The interpretation of the odds ratio had some errors and we accepted the comments from the reviewers. Hence, we thoroughly edited the suggestions in way that the Odds ratio could be interpreted accordingly. Hence, we interpreted it as increased occurrence or odds of IBG or central obesity as it is indicated in the revised version. The comments are well accepted.

Computing COR is not conclusive to determine determinants of IBG.

Here, we run both the crude and adjusted odds ratio. This could allow us to clearly indicate the statistical analysis and the reader in a very clear and reproducible way. Since we also clearly reported the adjusted association, this is very informative than the crude one. Hence, we made our conclusion and recommendations based on the adjusted associations. Presenting the crude one makes the paper more holistic and reproducible.

This discussion about the association between dietary pattern and central obesity and IBG in this study is not adequate. In addition, it is not exhaustive. You need to compare with similar finding and identify similar and unique variables. If this study identifies unique variable you need to have possible justification.

Here, as you mentioned the objective of the study is like that and we tried our best to present the discussion in a very clear and sequential manner. We started with IBG and central obesity, then their association with empirical dietary patterns. The major challenge was lack of such evidence in Ethiopia and Africa as well. That is why we mainly used gray literatures and fats to support and explain our results. But we have tried our best to modify the discussion as per the reviewers’ suggestions and authors’ view. These are indicated in revised manuscript. 

How do conclude that 94% had increased risk for cardiovascular complication? Because, you did not do any statistics test regarding risk for cardiovascular complication.

This is based on the WHO cutoff value for WHCR, which is predictive of the increased risk for cardiovascular complications. These has been indicated in many nutritional epidemiological studies and are very predictive for CVD risks.

Thanks for having a comment on the limitation of the study. Since mentioning the relevant limitation of the study is important to interpreted the study finding and its implication, we have mentioned and explained these limitations there.

Lastly, we have modified the recommendations to be specific to the specific body with a clear message. This is indicated in the conclusion part of the manuscript. 

.... higher among those who were physically active (AOR = 1.60; 95% CI: 0.90–2.85) (Table 5).

This result seems strange when we see the direct relationship between exercise and IBG. However, as studies reported and we tried to discuss in the discussion, majority of the exercisers usually have some sort of non-communicable disease and obesity. In such cases, the individual might have an already IBG despite exercise. We noted this result in more detail there.

Thank you very much!

Authors

---

## [Decision Letter · Decision Letter 1]

16 Feb 2023

PONE-D-22-28499R1Major dietary patterns of community dwelling adults and their associations with impaired blood glucose and central obesity in Eastern Ethiopia: diet-disease exploration studyPLOS ONE

Dear Dr. Oumer,

Thank you for submitting your manuscript to PLOS ONE. After careful consideration, we feel that it has merit but does not fully meet PLOS ONE’s publication criteria as it currently stands. Therefore, we invite you to submit a revised version of the manuscript that addresses the points raised during the review process.

We look forward to receiving your revised manuscript.

Kind regards,

Mohammad Asghari Jafarabadi

Academic Editor

PLOS ONE

Journal Requirements:

Reviewers' comments:

Reviewer's Responses to Questions

**Comments to the Author**

1. If the authors have adequately addressed your comments raised in a previous round of review and you feel that this manuscript is now acceptable for publication, you may indicate that here to bypass the “Comments to the Author” section, enter your conflict of interest statement in the “Confidential to Editor” section, and submit your "Accept" recommendation.

Reviewer #1: All comments have been addressed

Reviewer #2: (No Response)

2. Is the manuscript technically sound, and do the data support the conclusions?

Reviewer #1: Yes

Reviewer #2: Yes

3. Has the statistical analysis been performed appropriately and rigorously? 

Reviewer #1: Yes

Reviewer #2: Yes

4. Have the authors made all data underlying the findings in their manuscript fully available?

Reviewer #1: Yes

Reviewer #2: (No Response)

5. Is the manuscript presented in an intelligible fashion and written in standard English?

Reviewer #1: Yes

Reviewer #2: Yes

6. Review Comments to the Author

Reviewer #1: (No Response)

Reviewer #2: Computing COR is not conclusive to determine determinants of IBG.

Row 1 of the table is COR, which denotes for crude ODDS ratio. If it is adjusted, you must write AOR. Except that your justification to include crude odds ratio is acceptable. Line 381.

You need to scrutinize the interpretation of the ODDS result in the discussion section. There are couples of mistake there.

“Central obesity (AOR = 0.87: p-value 0.174)” Line 420 to 421, from where you brought this finding? Table 4 shows this finding “21.1(2.77-161.4)” the reference population is a physically active group.

ODDS ration interpretation. In the discussion and Abstract section. I need a clear-cut interpretation regarding ODDS ratio. If you did Hazard ratio, you can interpret as risk without hesitation.

7. PLOS authors have the option to publish the peer review history of their article (what does this mean?). If published, this will include your full peer review and any attached files.

Reviewer #1: No

Reviewer #2: No

---

## [Author Response · Author response to Decision Letter 1]

16 Feb 2023

Dear reviewer

Thanks for having your second-round revisions. We have corrected the comments as indicated in the revised version.

COR and AOR is distinctly different. 

Response: the COR and AOR are different and that is why we presented them for Central obesity and IBG separately in Table 3. Similarly, we clearly resented the AOR in the next table 4. The one in the table 3 is the unadjusted Bivariable logistic regression out put while the next table presented the adjusted associations.

Row 1 of the table is COR which Denotes for crude ODDS ratio. If it is Adjusted you must write AOR. Except that your justification to include crude odds ratio is acceptable.

Response: these was due to error and we have corrected this one in the revised version.

I need a clear-cut interpretation regarding ODDS ratio. If you did Hazard ratio you can interpret as risk without hesitation. 

Response: The major comment was ack of consistent interpretation of the odds ratio. Considering this, we have thoroughly addressed these interpretations in well-structured manner as likelihood rather than relative risk or hazard ratio.

Reviewer #2: Computing COR is not conclusive to determine determinants of IBG.

Row 1 of the table is COR, which denotes for crude ODDS ratio. If it is adjusted, you must write AOR. Except that your justification to include crude odds ratio is acceptable. Line 381.

You need to scrutinize the interpretation of the ODDS result in the discussion section. There are couples of mistake there.

Response: The major comment was ack of consistent interpretation of the odds ratio. Considering this, we have thoroughly addressed these interpretations in well-structured manner as likelihood rather than relative risk or hazard ratio.

“Central obesity (AOR = 0.87: p-value 0.174)” Line 420 to 421, from where you brought this finding? Table 4 shows this finding “21.1(2.77-161.4)” the reference population is a physically active group.

Response: as indicated this finding was obtained from ref no 46 as indicated there. 

ODDS ration interpretation. In the discussion and Abstract section. I need a clear-cut interpretation regarding ODDS ratio. If you did Hazard ratio, you can interpret as risk without hesitation.

Response: The major comment was ack of consistent interpretation of the odds ratio. Considering this, we have thoroughly addressed these interpretations in well-structured manner as likelihood rather than relative risk or hazard ratio.

---

## [Decision Letter · Decision Letter 2]

2 Mar 2023

Major dietary patterns of community dwelling adults and their associations with impaired blood glucose and central obesity in Eastern Ethiopia: diet-disease epidemiological study

PONE-D-22-28499R2

Dear Dr. Oumer,

We’re pleased to inform you that your manuscript has been judged scientifically suitable for publication and will be formally accepted for publication once it meets all outstanding technical requirements.

Kind regards,

Mohammad Asghari Jafarabadi

Academic Editor

PLOS ONE 

Reviewers' comments:

Reviewer's Responses to Questions

**Comments to the Author**

1. If the authors have adequately addressed your comments raised in a previous round of review and you feel that this manuscript is now acceptable for publication, you may indicate that here to bypass the “Comments to the Author” section, enter your conflict of interest statement in the “Confidential to Editor” section, and submit your "Accept" recommendation.

Reviewer #1: All comments have been addressed

Reviewer #2: All comments have been addressed

2. Is the manuscript technically sound, and do the data support the conclusions?

Reviewer #1: Yes

Reviewer #2: Yes

3. Has the statistical analysis been performed appropriately and rigorously? 

Reviewer #1: Yes

Reviewer #2: I Don't Know

4. Have the authors made all data underlying the findings in their manuscript fully available?

Reviewer #1: Yes

Reviewer #2: Yes

5. Is the manuscript presented in an intelligible fashion and written in standard English?

Reviewer #1: Yes

Reviewer #2: Yes

6. Review Comments to the Author

Reviewer #1: (No Response)

Reviewer #2: I have no comments and suggestion. I fully support the publication. The authors addressed all comments given to them. I really appreciate their effort and energy.

7. PLOS authors have the option to publish the peer review history of their article (what does this mean?). If published, this will include your full peer review and any attached files.

Reviewer #1: No

Reviewer #2: No

---

## [Editor Report · Acceptance letter]

11 Apr 2023

PONE-D-22-28499R2 

Major dietary patterns of community dwelling adults and their associations with impaired blood glucose and central obesity in Eastern Ethiopia: diet-disease epidemiological study 

Dear Dr. Oumer:

I'm pleased to inform you that your manuscript has been deemed suitable for publication in PLOS ONE. Congratulations! Your manuscript is now with our production department. 

Kind regards, 

on behalf of

Professor Mohammad Asghari Jafarabadi 

Academic Editor

PLOS ONE